# Examining the Role of Interpersonal Violence in Racial Disparities in Breastfeeding in North Dakota (ND PRAMS 2017–2019)

**DOI:** 10.3390/ijerph20085445

**Published:** 2023-04-09

**Authors:** MichaeLynn Kanichy (Makah), Lexie Schmidt, RaeAnn Anderson, Grace Njau, Amy Stiffarm (Aaniiih), Matthew Schmidt, Anastasia Stepanov, Andrew Williams

**Affiliations:** 1Public Health Program, Department of Population Health, School of Medicine & Health Sciences, The University of North Dakota, 1301 North Columbia Road Stop 9037, Grand Forks, ND 58202, USA; 2Department of Psychology, University of North Dakota, 501 North Columbia Road Stop 8380, Grand Forks, ND 58202, USA; 3North Dakota Department of Health & Human Services, 600 East Boulevard Ave, Department 325, Bismarck, ND 58505, USA; 4Department of Indigenous Health, School of Medicine & Health Sciences, University of North Dakota, 1301 North Columbia Road Stop 9037, Grand Forks, ND 58202, USA

**Keywords:** breastfeeding, breastfeeding initiation, breastfeeding duration, American Indian, racial disparities, interpersonal violence, violence, indigenous, pregnancy risk assessment monitoring system, North Dakota

## Abstract

Background. The 2019 overall breastfeeding initiation rate in the US was 84.1%, yet only 76.6% of American Indian (AI) women initiated breastfeeding. In North Dakota (ND), AI women have greater exposure to interpersonal violence than other racial/ethnic groups. Stress associated with interpersonal violence may interfere with processes important to breastfeeding. We explored whether interpersonal violence partially explains racial/ethnic disparities in breastfeeding in ND. Methods. Data for 2161 women were drawn from the 2017–2019 ND Pregnancy Risk Assessment Monitoring System. Breastfeeding questions in PRAMS have been tested among diverse populations. Breastfeeding initiation was self-report to “Did you ever breastfeed or pump breast milk to feed your new baby, even for a short period?” (yes/no). Breastfeeding duration (2 months; 6 months) was self-reported how many weeks or months of breastmilk feeding. Interpersonal violence for both 12 months before and during pregnancy based on self-report (yes/no) of violence from a husband/partner, family member, someone else, or ex-husband/partner. An “Any violence” variable was created if participants reported “yes” to any violence. Logistic regression models estimated crude and adjusted odds ratios (OR) and 95% confidence intervals (95% CI) for breastfeeding outcomes among AI and Other Race women compared to White women. Sequential models were adjusted for interpersonal violence (husband/partner, family member, someone else, ex-husband/partner, or any). Results. AI women had 45% reduced odds of initiating breastfeeding (OR: 0.55, 95% CI: 0.36, 0.82) compared to white women. Including interpersonal violence during pregnancy did not change results. Similar patterns were observed for all breastfeeding outcomes and all interpersonal violence exposures. Discussion. Interpersonal violence does not explain the disparity in breastfeeding in ND. Considering cultural ties to the tradition of breastfeeding and the role of colonization may provide a better understanding of breastfeeding among AI populations.

## 1. Introduction

Breastfeeding provides an array of health benefits for both babies (reduced risk of lower respiratory tract infections, sudden infant death syndrome, and neurodevelopmental conditions) and mothers (reduced risk of postpartum depression, ovarian and breast cancer, and type 2 diabetes) [1,2]. In addition, mothers who breastfeed their children for longer durations show more maternal sensitivity well past the infant and toddler years [3]. Breastfeeding in tribal communities is particularly important as chronic health conditions such as diabetes disproportionately impact American Indian populations compared to other races and ethnicities [4,5,6]. Not only is breastfeeding important because of its ability to provide health benefits, breastfeeding has been an important traditional practice across diverse American Indian populations [7,8]. Breastfeeding is a way Indigenous mothers can reclaim traditional practices by connecting with babies physically, emotionally, and spiritually. Furthermore, data suggest a strong association between prenatal use of traditional practices and breastfeeding at six months among Indigenous mothers in the Midwest region [9].

Despite the health and cultural-related benefits of breastfeeding for American Indian populations, persistent disparities exist. National data indicate that the breastfeeding initiation rate in 2019 was 84.1%, surpassing the Healthy People 2020 goal of 81.9% [10]. However, compared to women of other races and ethnicities, American Indian women and Black women had the lowest breastfeeding initiation rates (76.6% and 73.6%, respectively) [10]. North Dakota has a significant American Indian population, as American Indians are the second largest racial group (5.0%) after Whites [11]. North Dakota also has the largest disparity in 2019 breastfeeding initiation rates (37.6 percentage points) by race/ethnicity, with American Indian women reporting the lowest rates of breastfeeding initiation at 54.0%, compared to 91.6% among Asian women, 88.2% among White women, and 85% among all women in the state [10]. Thus, a closer examination of what impacts breastfeeding disparities in North Dakota is warranted.

Interpersonal violence is the intentional use of physical force or power against other persons or groups of persons, including physical, sexual, or psychological violence as well as deprivation and neglect. Interpersonal violence includes childhood maltreatment, family, and intimate partner violence (IPV), and community violence among non-related individuals [12]. Interpersonal violence has been linked to poorer breastfeeding outcomes, including less initiation and decreased duration, despite violence survivors often having greater intentions to begin breastfeeding [13,14,15]. The stress of being mistreated [16] appears to interfere biologically and psychologically with breastfeeding. Even for those not currently being abused, qualitative research suggests that for childhood sexual abuse survivors, breastfeeding can activate body shame, feelings of lack of control of their bodies, and concerns around sexuality [17]. Violence is a public health crisis in American Indian communities. Data indicates that 4 in 5 American Indian women (84.3%) have experienced sexual violence in their lifetime [18].

Furthermore, violence on reservations has been reported to be ten times higher than the national average [19]. IPV is the most common form of interpersonal violence committed against women, and American Indian women disproportionately experience IPV at a rate 1.6 times higher than non-Hispanic White women [18]. Violence against pregnant women is also concerning, as homicide, which is greatly associated with IPV, is the number one cause of death among pregnant and postpartum women [20]. In addition, 8.1% of women experience IPV 12 months before pregnancy, and 4.7% of women experience IPV during pregnancy [21]. In North Dakota, approximately 30% of women experience IPV in their lifetimes [22], and at least 3% of new victims were pregnant at the time of assault [23]. Exposure to violence is associated with high-stress levels and depression [24], which may influence breastfeeding initiation and duration [25]. Social support from healthcare professionals and families is also an important motivator for new mothers to breastfeed [9,26]. However, women who experience IPV may not have access to familial and social support.

To our knowledge, five studies have examined the relationship between IPV and breastfeeding outcomes. A cross-sectional study of 217 mothers from Rio de Janeiro found that households where couples physically abuse each other were less likely to breastfeed when compared to households without physical violence [27]. A cohort study including 52,000 women and their infants in Australia reported that IPV was associated with lower odds of breastfeeding initiation and breastfeeding six weeks postpartum [28]. Three studies assessed this association in the U.S. A prospective study of 101 pregnant Women, Infants, and Children (WIC)-enrolled women in Indiana found that IPV exposure 12 months before pregnancy was not associated with breastfeeding initiation. However, IPV during pregnancy was associated with lower odds of breastfeeding at six weeks postpartum [29]. An analysis of 2000–2003 National PRAMS data did not find significant associations between experiences of IPV and breastfeeding initiation nor breastfeeding four weeks postpartum [30]. However, an analysis of the 2004–2014 National PRAMS reported that women who experienced IPV 12 months before pregnancy were more likely to breastfeed at eight weeks postpartum than women who did not report IPV. However, they did not find a similar association between IPV during pregnancy and breastfeeding duration [25].

Overall, literature examining the association between interpersonal violence before and during pregnancy and breastfeeding initiation and duration has reported inconsistent results. Additionally, the association has not been solely examined among American Indian women. Therefore, this study examines whether exposure to interpersonal violence partially accounts for racial disparities in breastfeeding among American Indian and Alaskan Native women.

## 2. Materials and Methods

We used data from the North Dakota Pregnancy Risk Assessment Monitoring System (ND PRAMS) for 2017–2019. The ND PRAMS is conducted by the North Dakota Department of Health in collaboration with the Centers for Disease Control and Prevention (CDC). PRAMS is an ongoing, site-specific, population-based surveillance system designed to monitor maternal behaviors and experiences before, during and shortly after pregnancy [31]. During data collection, the ND PRAMS survey consisted of 103 multiple-choice and fill-in-the-blank questions regarding sociodemographic factors, maternal behaviors and experiences, and clinical outcomes. In addition, data from birth certificates are linked to ND PRAMS data to supplement the survey results. The CDC developed PRAMS survey questions and made them available as part of a “standard list” that states choose to include in their survey. These questions have been tested among diverse populations and have undergone assessment as part of CDC protocol. The survey methodology is the same for all racial/ethnic groups sampled.

Every month a sample of mothers is selected from live birth certificates filed in the previous month. In North Dakota, American Indian women are oversampled. The CDC oversamples American Indian women to ensure adequate data are available in smaller, higher-risk populations [31]. During the study period (2017–2019), mothers identifying as AIAN or AIAN/White represented 10.1% of total births (3199 of 31,813). ND PRAMS sampled 100% of women who identified as American Indian or biracial American Indian/White to participate in the ND PRAMS survey. Women who did not identify as American Indian were randomly sampled at a 6% rate. Birth rates among the groups included in the racial/ethnic groups were too low during the study period (2017–2019) to inform sampling strata and specific racial/ethnic variables for analyses. For this analysis, weighting accounts for oversampling, nonresponse, and noncoverage adjustments.

The ND PRAMS 2017–2019 sample included 4297 women (Figure 1). Women with missing data on the question, “Did you ever breastfeed or pump breast milk to feed your new baby, even for a short period?” were excluded (*n* missing = 2136) for a final study sample of 2161 women. Of the women excluded from the study, the majority were American Indian (62.50%), had an education level below high school (53.70%), and were not married (69.71%). After applying survey weights, the analytic sample was 29,845 women. Thus, each ND PRAMS participant represents approximately 13 women who recently gave birth in ND.

### 2.1. Race/Ethnicity

ND PRAMS participants self-reported race/ethnicity: American Indian (AI; AI alone or biracial AI-white), White (women who self-reported as white alone) and women of Other Racial Identities (any other race/ethnicity including Black, Asian, Hispanic). However, birth rates among the groups included in the racial/ethnic groups were too low during the study period (2017–2019) to inform sampling strata and specific racial/ethnic variables for analyses.

### 2.2. Breastfeeding

Breastfeeding initiation was measured by analyzing maternal responses to “Did you ever breastfeed or pump breast milk to feed your new baby, even for a short period?” (yes/no). Breastfeeding duration was measured by analyzing the maternal response to “Are you currently breastfeeding or feeding pumped milk to your new baby?” (yes/no) and “How many weeks or months did you breastfeed or feed pumped milk to your baby?” Mothers could respond by selecting “Less than 1 week” or reporting breastfeeding duration in weeks or months. Breastfeeding duration was identified as two months and six months (both variables, yes/no).

### 2.3. Interpersonal Violence

Interpersonal violence questions in PRAMS focus on physical violence. Violence experiences were measured using the mother’s response to, “During the 12 months before you got pregnant, did any of the following people push, hit, slap, kick, choke, or physically hurt you in any other way?” Mothers could report violence from a husband/partner, ex-husband/ex-partner, another family member, or someone else. An “Any exposure to violence” variable was created if women reported “yes” to any of the exposure types (husband/partner, ex-husband/ex-partner, another family member, or someone else). Violence exposures were obtained for 12 months before pregnancy and for during pregnancy.

### 2.4. Statistical Analysis

Descriptive statistics were summarized by race/ethnicity. A series of logistic regression models were fit to estimate odds ratios and 95% confidence intervals for breastfeeding outcomes among American Indian and Other Racial Identity women compared to White women. The first model estimated the crude association between race/ethnicity and breastfeeding outcomes. Next, maternal sociodemographic and medical factors were added to the crude mode. Then, in separate models, we included violence exposure for both 12 months before pregnancy and during pregnancy to assess whether the exposure accounted for observed racial/ethnic disparities.

Models were adjusted for maternal age (less than 20 years old, 21–35 years old, and greater than 35 years old; self-report), income (less than $48,000 and greater than $48,000; self-report), education (less than high school and greater than high school; derived from birth certificate), participation in the Women, Infants, and Children Nutrition (WIC; derived from birth certificate) program (yes, no), insurance prior to pregnancy (yes, no; self-report), pregnancy intention (yes, no; self-report), Kotelchuck Adequacy of Prenatal Care Utilization (APNCU; derived from birth certificate data) index (inadequate, intermediate, adequate, adequate plus) [32], Body Mass Index (less than 25, greater than 25; self-report), substance use during pregnancy (cigarette, marijuana, and e-cigarette use; self-report), marital status (yes, no; derived from birth certificate), presence of chronic illness and conditions (diabetes, high blood pressure, and depression; self-report), and adverse childhood experiences (ACEs) (less than 4, 4 or more; self-report). Covariates were informed by prior literature [29,33,34]. 

To determine if income or education may modify potential racial/ethnic disparities in breastfeeding, we performed sensitivity analyses by fitting two additional regression models for each breastfeeding outcome: (1) Excluding education from the fully adjusted model; (2) excluding income from the fully adjusted model.

Statistical analyses were conducted with SAS OnDemand for Academics (SAS Institute Inc., Cary, NC, USA) using proc survey commands and survey weights to account for complex survey design and nonresponse.

## 3. Results

Rates of breastfeeding initiation and covariates for the overall sample by race/ethnicity are in Table 1. Compared to white women, American Indian women and women of Other Racial Identities initiate breastfeeding at lower rates (90.19%, 68.87%, 88.94%, respectively) and experience higher rates of any interpersonal violence before pregnancy (11.68%, 2.75%, 1.54% respectively), and during pregnancy (7.16%, 1.70%, 1.72% respectively). American Indian women and women of Other Racial Identities were also more likely not to be married, have less than a high school education, and have an income below $48,000 than White women. American Indian women were also more likely to be younger than women of Other Racial Identities and White women. Table 2 includes the prevalence of exposure to interpersonal violence by race/ethnicity. American Indian women have the highest rates of self-reported exposure to interpersonal violence for all interpersonal violence variables analyzed.

Results for regression analyses are in Table 3, Table 4 and Table 5. Table 3 includes regression analysis results for the association between race/ethnicity, breastfeeding initiation, and violence before and during pregnancy. In the crude model, American Indian women had 76% significantly lower odds of initiating breastfeeding (OR: 0.24, 95% CI: 0.18, 0.31) than White women. Adjusting for covariates accounted for 41% of the disparity. However, American Indian women were still 45% less likely to initiate breastfeeding (OR: 0.55, 95% CI: 0.37, 0.83). Upon adjusting for “any exposure to violence” before pregnancy, the odds for breastfeeding initiation among American Indian women did not change (OR: 0.55, 95% CI: 0.36, 0.82). Adjusting for “any exposure to violence” during pregnancy also did not alter the odds of breastfeeding initiation (OR: 0.55, 95% CI: 0.37, 0.83).

Table 4 includes regression results for the association between race/ethnicity, breastfeeding for at least two months, and violence before and during pregnancy. In the crude model, American Indian women were 70% significantly less likely (OR: 0.30, 95% CI: 0.25, 0.36) to breastfeed for at least two months compared to White women. Adjusting for covariates reduced the disparity by 57%, but American Indian women still had 30% significantly lower odds (OR: 0.70, 95% CI: 0.53, 0.94) of breastfeeding for at least two months than White women. In addition, separate adjustments for any violence both before and during pregnancy did not significantly change the likelihood of American Indian women breastfeeding for at least two months (Before—OR: 0.70, 95% CI: 0.53, 0.94; During—OR: 0.71, 95% CI: 0.53, 0.95).

Table 5 contains regression results for the association between race/ethnicity, breastfeeding for at least six months, and violence before and during pregnancy. In the crude model, American Indian women were 76% significantly less likely (OR: 0.24, 95% CI: 0.20, 0.29) to breastfeed for at least six months compared to White women. Adjusting for covariates reduced the disparity by 50%, but American Indian women still had 38% significantly lower odds (OR: 0.62, 95% CI: 0.47, 0.83) of breastfeeding for at least six months than White women. In addition, separate adjustments for any violence both before and during pregnancy did not significantly change the likelihood of American Indian women breastfeeding for at least six months (Before—OR: 0.62, 95% CI: 0.47, 0.83; During—OR: 0.62, 95% CI: 0.47, 0.84).

The results of the sensitivity analyses do not differ from our main analyses, suggesting income and education do not significantly influence observed racial/ethnic disparities in breastfeeding (Appendix A).

## 4. Discussion

In this analysis of the role of interpersonal violence in breastfeeding disparities in North Dakota, we hypothesized that exposure to interpersonal violence would partially account for disparities in breastfeeding initiation and duration among American Indian women. However, accounting for interpersonal violence before or during pregnancy did not mitigate racial disparities in breastfeeding outcomes. As noted in the Normann et al. 2020 systematic review [15], interpersonal violence is generally associated with poorer breastfeeding outcomes. Yet, the relationship between interpersonal violence and breastfeeding is complex and accounting for different covariates changes findings.

The observation of American Indian women being 60% less likely to initiate breastfeeding aligns with observations from ND vital statistics data. An ecologic analysis of ND vital statistics data found that American Indian women were 2.7 times less likely to initiate breastfeeding than white women [35]. However, ecologic study designs are limited by their lack of consideration of individual-level factors. The present study allowed for control of individual-level covariates of breastfeeding initiation and duration. The present study suggests that individual-level factors (i.e., socioeconomic factors, health conditions, insurance coverage) account for approximately 60% of the disparity in breastfeeding initiation and 50–60% in breastfeeding duration. Despite accounting for a range of individual-level covariates, including exposure to violence, American Indian women are 45% less likely to initiate breastfeeding and 30–40% less likely to meet breastfeeding duration targets.

Our findings suggest interpersonal violence, as measured in ND PRAMS, does not partially account for breastfeeding disparities, which aligns with other cross-sectional PRAMS studies. For example, the Silverman et al. [30] study found that women who experienced IPV during pregnancy were significantly (OR: 1.35, CI 1.26–1.66) more likely not to breastfeed. However, after adjusting for similar individual-level covariates, including age, race, marital status, education, receipt of governmental assistance, and current smoking, women who reported experiencing IPV during pregnancy were not at an increased risk for not breastfeeding [30]. Although Silverman utilized PRAMS (2000–2003) and adjusted for similar individual-level covariates, the current study used PRAMS data site-specific to North Dakota, where American Indian women are oversampled. Although the literature on the association between breastfeeding and IPV is limited in the United States, the current study is one of the only studies focusing on American Indian mothers.

While our findings align with findings from Silverman [30], other U.S. studies suggested that IPV may reduce the likelihood of breastfeeding initiation and duration [35,36]. Prior literature found that women reporting prenatal IPV were significantly (OR: 0.22 CI: 0.05, 0.85) less likely to be breastfeeding at 6-week postpartum follow-up [36]. Breastfeeding initiation was measured using the same PRAMS variable as the current study. However, differences in our findings could be attributed to study design, particularly violence measurement and sample effects. Miller-Graff et al. conducted a prospective longitudinal survey following 69 low-income women receiving services from a local Women, Infants and Children (WIC) program in Indiana [36]. Although nearly 38% of the sample reported IPV, none of the participants was American Indian in this study. The Miller-Graff et al. study also controlled for biopsychosocial risk and protective factors, including pregnancy-related health problems and depressed mood. Although the Miller-Graff et al. study could not offer any precise, mechanistic explanations as to why IPV might have a negative impact on breastfeeding, many of the factors theorized by Miller-Graff and colleagues, such as adversity and perinatal health, would influence American Indian women.

When addressing racial health inequities, including historical context and the implications is important. The impact structural racism, in the form of colonization and subsequent federal policies, has had on breastfeeding behavior among American Indian populations is an additional barrier that is difficult to measure. For example, by the turn of the twentieth century, American Indian populations were forcibly removed from their ancestral lands and relocated to reservations. High rates of disease and illness, including high infant mortality rates, triggered the Office of Indian Affairs to launch the Save the Babies campaign. Part of the Save the Babies campaign included the field matrons program sending hundreds of Euro-American women to teach American Indian women “scientific motherhood” [37]. As a result, field staff on many reservations began to discourage breastfeeding. They presented traditional breastfeeding practices of American Indian communities as barbaric and backwards and promoted cow’s milk as a more scientifically attractive mode of feeding infants [37]. However, the promotion of cow’s milk was quickly reversed due to the lack of access to it and adequate sanitation in its preparation [37]. Today, American Indian mothers continue to feel the ripple effect of federal policies that replaced traditional breastfeeding practices (e.g., community lactation consultants) with inaccessible Western breastfeeding practices. Among Navajo mothers, “breastfeeding difficulties including pain, latch issues, and concerns about inadequate milk supply, especially in women who breastfed for six months or less. Most women felt they could have benefitted from lactation consultant support, which studies suggest may ameliorate breastfeeding problems and improve breastfeeding outcomes” [34].

Another factor not measured in the PRAMS data was whether women were the primary breadwinners for their households. A breadwinner mother is defined as a single mother who heads a household or a married mother who earns at least half the couple’s joint income [38]. The Institute for Women’s Policy Research published a study that indicated approximately 2 out of 3 American Indian mothers are primary breadwinners, most of whom are single mothers [38]. The PRAMS analysis is in line with this finding in which American Indian women were significantly less likely to be married in comparison to White women and women of Other Racial Identities (18.53%, 71.82%, and 67.46%). Furthermore, American Indian mothers are more likely to be experiencing economic hardship, where approximately 25% of female-headed households were in poverty compared to 4.7% of married couples [38]. Perhaps American Indian mothers who hold the burden of maintaining the household find it difficult to find the time and energy to breastfeed their new infants. Professionals recommend feeding every 2–3 h for 20–30 min, taking up anywhere to 2–6 h of the day [38].

The results of this analysis should be considered in the context of its strengths and limitations. To our knowledge, this is the first study to examine whether interpersonal violence accounts for racial/ethnic disparities in breastfeeding, specifically among American Indian women. As American Indian populations have high rates of interpersonal violence, and low rates of breastfeeding, these data are important to understanding maternal and infant health among American Indian populations. Next, the weighted design used in this study oversampled American Indian women, allowing the sample analysis better to represent this population of pregnant women in ND. The greater inclusion of American Indian women is important for ameliorating maternal/infant health disparities and centering American Indian culture.

Additionally, our approach using individual-level observations from ND PRAMS data allowed for control of individual-level factors, as the prior ecologic study of prenatal care initiation in ND did not account for individual-level factors in the analysis. This study does have several limitations. First, this is a cross-sectional design, limiting causal interpretations. However, data on interpersonal violence reflect occurrences before and during pregnancy. Thus, the data shows a temporal relationship between interpersonal violence and breastfeeding. Second, those excluded from analysis for missing variables were 62.5% AI/AN and had an education level below high school (53.70%), thus likely being high risk for not initiating breastfeeding and having short breastfeeding duration, as well as being high risk for the experience of interpersonal violence. Therefore, the estimated disparity in breastfeeding outcomes, and the observed lack of effect attributed to interpersonal violence, can be considered conservative. Third, PRAMS only collects one yes/no question on breastfeeding initiation and does not collect data on exclusive breastfeeding or mixed feedings. Thus, our observations do not apply to mixed feeding, alternative nutrition, or the introduction of solid foods for infants. Fourth, the measurement of interpersonal violence was limited to a single question with four potential responses in PRAMS data. Research on the measurement of violence suggests that the more items on violence administered the higher prevalence rates reported [39]. Fifth, the findings among the other racial/ethnic group should be interpreted with caution, given the heterogeneity of that group. However, given this heterogeneity, the results suggest women within this group may have similar rates of breastfeeding initiation and breastfeeding as White women in ND.

## 5. Conclusions

Our results highlight that breastfeeding disparities exist among American Indian women, and prevalent risk factors such as interpersonal violence do not explain observed disparities. Culturally responsive interventions (e.g., breastfeeding education and lactation support) are needed to improve breastfeeding outcomes among American Indian mothers, yet research supporting culturally responsive interventions is sparse within AI populations. Further research should clarify unique socioeconomic circumstances, such as the women’s role as a breadwinner and their impact on breastfeeding disparities. As a historically systemically oppressed population in which health policies have diminished cultural practices in the past, ethical research practices, including data sovereignty, data ownership, and publications, should be at the respective tribe’s discretion in relation to any future research.

## Figures and Tables

**Figure 1 ijerph-20-05445-f001:**
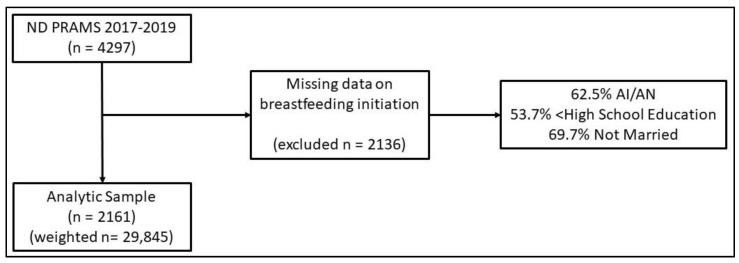
Study inclusion/exclusion flow chart.

**Table 1 ijerph-20-05445-t001:** Weighted Percent (and frequency) of breastfeeding outcomes and covariates, overall and by race/ethnicity in ND PRAMS (2017–2019) *.

	Overall (*n* = 29,845)	American Indian (*n* = 2499)	White (*n* = 23,585)	Other Racial Identities (*n* = 3760)
Initiation of breastfeeding ^a^				
Yes	88.24 (26,336)	68.87 (1721)	90.19 (21,271)	88.94 (3344)
No	11.76 (3509)	31.13 (778)	9.81 (2314)	11.06 (416)
Breastfed at least 2 mo. ^a^				
Yes	67.28 (20,079)	40.90 (1022)	69.45 (16,379)	71.22 (2678)
No	32.72 (9765)	59.10 (1477)	30.55 (7206)	28.78 (1082)
Breastfed at least 6 mo. ^a^				
Yes	59.08 (17,633)	28.81 (720)	62.27 (14,687)	59.18 (2225)
No	40.92 (12,212)	71.19 (1779)	37.73 (8898)	40.82 (1535)
Maternal Age in years ^a^				
<=20	3.40 (1016)	12.41 (310)	2.68 (632)	1.97 (74)
21-35	83.55 (24,934)	77.75 (1943)	85.43 (20,148)	75.64 (2844)
>35	13.05 (3894)	9.84 (246)	11.89 (2805)	22.39 (842)
Income ^a^				
$48,001 or above	60.29 (17,995)	14.97 (374)	70.21 (16,560)	28.19 (1060)
$48,000 or below	39.71 (11,850)	85.03 (2125)	29.79 (7025)	71.81 (2700)
Insurance Before Pregnancy ^a^				
Yes	92.46 (27,593)	92.00 (2299)	94.00 (22,168)	83.14 (3126)
No	7.54 (2251)	8.00 (200)	6.00 (1417)	16.86 (634)
Education Level ^a^				
More than HS	70.26 (20,968)	44.46 (1111)	75.91 (17,903)	51.97 (1954)
Less than or Equal to HS	29.74 (8876)	55.54 (1388)	24.09 (5682)	48.03 (1806)
Received WIC ^a^				
Yes	23.5 (7030)	55.8 (1396)	15.6 (3679)	51.9 (1955)
No	76.5 (22815)	44.2 (1103)	84.4 (19,906)	48.1 (1805)
Substance Use During Pregnancy ^a^				
Yes	3.76 (1121)	9.64 (241)	3.16 (746)	3.54 (133)
No	96.24(28,723)	90.36 (2258)	96.84 (22,839)	96.46 (3627)
Pregnancy Intention				
Did not want	5.31 (1584)	7.56 (189)	4.84 (1141)	6.76 (254)
Wanted	94.69 (28,261)	92.44 (2310)	95.16 (22,444)	93.24 (3506)
Kotelchuck index ^a^				
Inadequate	12.3 (3669)	43.5 (1089)	8.3 (1967)	16.3 (613)
Intermediate	10.6 (3180)	15.4 (384)	8.9 (2099)	18.5 (697)
Adequate	54.3 (16,193)	26.9 (671)	58.3 (13,740)	47.4 (1782)
Adequate Plus	22.8 (6802)	14.2 (355)	24.5 (5779)	17.8 (668)
Chronic Disease ^a^				
Yes	20.08 (5925)	27.17 (673)	20.78 (4867)	10.70 (386)
No	79.92 (23586)	72.83 (1804)	79.22 (18558)	89.30 (3223)
Overweight ^a^				
Yes	53.06 (15,835)	62.30 (1557)	53.21 (12,548)	46.00 (1730)
No	46.94 (14,010)	37.70 (942)	46.79 (11,036)	54.00 (2031)
Marital Status ^a^				
Yes	66.81 (19,940)	18.53 (463)	71.82 (16,939)	67.46 (2537)
No	33.19 (9905)	81.47 (2036)	28.18 (6646)	32.54 (1224)
High ACEs ^a^				
Yes	18.65 (5565)	29.73 (743)	19.11 (4506)	8.35 (314)
No	81.35 (24,280)	70.27 (1756)	80.89 (19,078)	91.65 (3446)

* *p*-values for racial/ethnic differences obtained with Rao-Scott chi-square in proc surveyfreq. ^a^ *p*-value for racial/ethnic differences < 0.01.

**Table 2 ijerph-20-05445-t002:** Weighted Relative (and absolute) frequency of violence exposure by race/ethnicity in ND PRAMS ^1^.

	Overall (*n* = 29,845)	American Indian (*n* = 2499)	White (*n* = 23,585)	Other Racial Identities (*n* = 3760)
Any Violence Exposure Before Pregnancy ^a^				
Yes	3.35 (1000)	11.68 (292)	2.75 (649)	1.54 (58)
No	96.65 (28,845)	88.32 (2207)	97.24 (22,936)	98.46 (3702)
Husband/Partner ^a^				
Yes	2.07 (618)	7.64 (191)	1.48 (349)	2.05 (77)
No	97.93 (29,227)	92.36 (2308)	98.52 (23,236)	97.95 (3683)
Ex-Husband/Partner ^a^				
Yes	2.12 (632)	5.60 (140)	1.69 (399)	2.45 (92)
No	97.88 (29,213)	94.40 (2359)	98.31 (23,186)	97.55 (3668)
Family ^a^				
Yes	1.50 (447)	5.76 (144)	0.94 (221)	2.15 (81)
No	98.50 (29,398)	94.24 (2355)	99.06 (23,364)	97.85 (3679)
Other				
Yes	1.93 (575)	4.04 (101)	1.77 (418)	1.46 (55)
No	98.07 (29270)	95.96 (2398)	98.23 (23167)	98.54 (3705)
Any Violence Exposure During Pregnancy				
Yes	2.17 (647)	7.16 (179)	1.70 (403)	1.72 (65)
No	97.83 (29,198)	92.84 (2320)	98.30 (23,182)	98.28 (3695)
Husband/Partner ^a^				
Yes	2.01 (601)	6.20 (155)	1.42 (336)	2.90 (109)
No	97.99 (29,244)	93.80 (2344)	98.58 (23,249)	97.10 (3651)
Ex-Husband/Partner ^a^				
Yes	1.45 (433)	4.32 (108)	1.12 (264)	1.59 (60)
No	98.55 (29,412)	95.68 (2391)	98.88 (23,321)	98.40 (3700)
Family ^a^				
Yes	1.47 (438)	5.80 (145)	0.94 (222)	1.84 (69)
No	98.53 (29,407)	94.20 (2354)	99.06 (23,363)	98.16 (3691)
Other				
Yes	1.57 (468)	3.40 (85)	1.44 (340)	1.14 (43)
No	98.43 (29,377)	96.60 (2414)	98.56 (23,245)	98.86 (3717)

^1^*p*-values for racial/ethnic differences obtained with Rao-Scott chi-square in proc surveyfreq. ^a^ *p*-value for racial/ethnic differences < 0.05.

**Table 3 ijerph-20-05445-t003:** Association between race/ethnicity and breastfeeding initiation, adjusted for violence exposure before and during pregnancy ^1^.

		Breastfeeding Initiation OR (95% CI)			
		American Indian	Other Racial Identities	White	Violence Variable Estimate
Crude Model		0.24 (0.18, 0.31) ^a^	0.87 (0.49, 1.52)	Ref.	-
Sociodemographic Model		0.55 (0.37, 0.83) ^a^	1.14 (0.60, 2.18)	Ref.	-
Violence Before Pregnancy	Any Interpersonal Violence	0.55 (0.36, 0.82) ^a^	1.16 (0.61, 2.22)	Ref.	0.39 (0.11, 1.42)
	Husband/Partner	0.55 (0.37, 0.83) ^a^	1.14 (0.60, 2.18)	Ref.	0.94 (0.37, 2.36)
	Ex- Husband/Partner	0.55 (0.37, 0.83) ^a^	1.14 (0.60, 2.18)	Ref.	0.95 (0.36, 2.18)
	Other Family	0.56 (0.37, 0.84) ^a^	1.15 (0.60, 2.19)	Ref.	0.68 (0.24, 1.95)
	Other	0.55 (0.37, 0.83) ^a^	1.15 (0.60, 2.19)	Ref.	0.49 (0.18, 1.31)
Violence During Pregnancy	Any Interpersonal Violence	0.55 (0.37, 0.83) ^a^	1.14 (0.60, 2.18)	Ref.	0.77 (0.20, 2.89)
	Husband/Partner	0.54 (0.36, 0.82) ^a^	1.14 (0.59, 2.17)	Ref.	1.33 (0.52, 3.42)
	Ex- Husband/Partner	0.55 (0.37, 0.83) ^a^	1.14 (0.60, 2.18)	Ref.	1.31 (0.46, 3.73)
	Other Family	0.55 (0.37, 0.83) ^a^	1.14 (0.60, 2.18)	Ref.	0.87 (0.34, 2.23)
	Other	0.55 (0.37, 0.83) ^a^	1.14 (0.60, 2.18)	Ref.	1.25 (0.42, 3.67)

^1^ Adjusted for race, income, insurance before pregnancy, maternal age, education, substance use, marital status, pregnancy intention, chronic disease, overweight/obese, WIC, Kotelchuck, and ACEs. ^a^ *p*-value < 0.05.

**Table 4 ijerph-20-05445-t004:** Association between race/ethnicity and breastfeeding for at least two months, adjusted for violence exposure before and during pregnancy ^1^.

		Breastfeeding Duration—2 Months OR (95% CI)			
		American Indian	Other Racial Identities	White	Violence Variable Estimate
Crude Model		0.30 (0.25, 0.36) ^a^	1.08 (0.75, 1.57)	Ref.	-
Sociodemographic Model		0.70 (0.53, 0.94) ^a^	1.25 (0.77, 2.03)	Ref.	-
Violence Before Pregnancy	Any Interpersonal Violence	0.70 (0.53, 0.94) ^a^	1.27 (0.78, 2.05)	Ref.	0.36 (0.08, 1.60)
	Husband/Partner	0.72 (0.54, 0.97) ^a^	1.27 (0.78, 2.05)	Ref.	0.59 (0.23, 1.49)
	Ex- Husband/Partner	0.71 (0.53, 0.94) ^a^	1.26 (0.77, 2.04)	Ref.	0.69 (0.29, 1.62)
	Other Family	0.72 (0.54, 0.96) ^a^	1.26 (0.78, 2.05)	Ref.	0.44 (0.13, 1.44)
	Other	0.70 (0.53, 0.94) ^a^	1.25 (0.77, 2.03)	Ref.	0.48 (0.18, 1.21)
Violence During Pregnancy	Any Interpersonal Violence	0.71 (0.53, 0.95) ^a^	1.27 (0.78, 2.06)	Ref.	0.23 (0.05, 1.09)
	Husband/Partner	0.72 (0.54, 0.97) ^a^	1.28 (0.79, 2.07)	Ref.	0.49 (0.19, 1.21)
	Ex- Husband/Partner	0.71 (0.53, 0.95) ^a^	1.26 (0.78, 2.04)	Ref.	0.45(0.17, 1.17)
	Other Family	0.72 (0.54, 0.96) ^a^	1.26 (0.78, 2.04)	Ref.	0.50 (0.16, 1.53)
	Other	0.71 (0.53, 0.95) ^a^	1.25 (0.78, 2.04)	Ref.	0.48 (0.18, 1.21)

^1^ Adjusted for race, income, insurance before pregnancy, maternal age, education, substance use, marital status, pregnancy intention, chronic disease, overweight/obese, WIC, Kotelchuck, and ACEs. ^a^ *p*-value < 0.05.

**Table 5 ijerph-20-05445-t005:** Association between race/ethnicity and breastfeeding for at least six months, adjusted for violence exposure before and during pregnancy ^1^.

		Breastfeeding Duration—6 Months OR (95% CI)			
		American Indian	Other Racial Identities	White	Violence Variable Estimate
Crude Model		0.24 (0.20, 0.29)	0.87 (0.62, 1.23)	Ref.	-
Sociodemographic Model		0.62 (0.47, 0.83)	0.89 (0.57, 1.38)	Ref.	-
Violence Before Pregnancy	Any Interpersonal Violence	0.62 (0.47, 0.83)	0.89 (0.57, 1.39)	Ref.	0.61 (0.11, 3.13)
	Husband/Partner	0.63 (0.47, 0.85)	0.89 (0.57, 1.39)	Ref.	0.67 (0.24, 1.90)
	Ex- Husband/Partner	0.62 (0.47, 0.83)	0.89 (0.57, 1.39)	Ref.	0.46 (0.18, 1.16)
	Other Family	0.63 (0.48, 0.85)	0.89 (0.57, 1.39)	Ref.	0.47 (0.12, 1.78)
	Other	0.62 (0.47, 0.83)	0.89 (0.57, 1.39)	Ref.	0.70 (0.24, 2.00)
Violence During Pregnancy	Any Interpersonal Violence	0.62 (0.47, 0.84)	0.90 (0.58, 1.40)	Ref.	0.34 (0.06, 1.88)
	Husband/Partner	0.64 (0.48, 0.85)	0.90 (0.58, 1.41)	Ref.	0.51 (0.19, 1.36)
	Ex- Husband/Partner	0.62 (0.47, 0.83)	0.89 (0.57, 1.39)	Ref.	0.70 (0.24, 1.98)
	Other Family	0.63 (0.47, 0.84)	0.89 (0.57, 1.39)	Ref.	0.70 (0.20, 2.46)
	Other	0.62 (0.47, 0.83)	0.89 (0.57, 1.39)	Ref.	0.63 (0.22, 1.77)

^1^ Adjusted for race, income, insurance before pregnancy, maternal age, education, substance use, marital status, pregnancy intention, chronic disease, overweight/obese, WIC, Kotelchuck, and ACEs.

## Data Availability

North Dakota PRAMS data are available after project approval from the North Dakota Department of Health and Human Services: https://www.hhs.nd.gov/prams (accessed on 15 October 2021).

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
