# Peer review of "Examining the Role of Interpersonal Violence in Racial Disparities in Breastfeeding in North Dakota (ND PRAMS 2017–2019)"

_ijerph, 2023, doi:10.3390/ijerph20085445_

Round 1

Reviewer 1 Report

Starting with interpersonal violence, the authors try to explains racial/ethnic disparities in breastfeeding among American Indian women. This produces far-reaching significance for improving breastfeeding in the population. But authors still need to solve some problems as follows.

1. The title is not clear enough. The main content is to demonstrate the unsatisfactory breastfeeding between different Race/ethnic group whether caused by interpersonal violence, not the juxtaposition of “Interpersonal Violence and Racial Disparities ”in title.

2. “Breastfeeding in tribal communities is particularly important as chronic health conditions such as obesity and diabetes disproportionately impact American Indian populations compared to other races and ethnicities“Does this mean breastfeeding reduces chronic health conditions such as obesity and diabetes in mothers? If yes, please explain the relationship between breastfeeding and maternal disease first.

3. A total of 2136 mothers missed data on the question and were excluded. Will more than half the number of excluded population lead to result deviation? The authors needs to explain it.

4. The authors should supplement the inclusion flow chart to make the inclusion steps clearer.

5.About statistics, the authors should supplement the P value between the two groups in Table 1 and Table 2 (different letter superscripts can be used). About OR value in the article,please add specific P value in corresponding description and table

Author Response

Reviewer #1:

Starting with interpersonal violence, the authors try to explains racial/ethnic disparities in breastfeeding among American Indian women. This produces far-reaching significance for improving breastfeeding in the population. But authors still need to solve some problems as follows.

  1. The title is not clear enough. The main content is to demonstrate the unsatisfactory breastfeeding between different Race/ethnic group whether caused by interpersonal violence, not the juxtaposition of “Interpersonal Violence and Racial Disparities ”in title.

Thank you for this suggestion. We have edited the title to be “Examining the role of Interpersonal Violence in Racial Disparities in Breastfeeding in North Dakota (ND PRAMS 2017-2019).”

  1. “Breastfeeding in tribal communities is particularly important as chronic health conditions such as obesity and diabetes disproportionately impact American Indian populations compared to other races and ethnicities“Does this mean breastfeeding reduces chronic health conditions such as obesity and diabetes in mothers? If yes, please explain the relationship between breastfeeding and maternal disease first.

Thank you for this comment. Upon additional review of the literature, we have removed the mention of obesity, as the evidence is weak that maternal breastfeeding reduces risk for maternal obesity later in life. However, the evidence is sufficient to suggest that maternal breastfeeding reduces maternal risk for type 2 diabetes in the future. We have revise the text in the introduction to explain that breastfeeding is associated with lower risk of type 2 diabetes after pregnancy (page 1, line 44):

Breastfeeding provides an array of health benefits for both babies (reduced risk of lower respiratory tract infections, sudden infant death syndrome, and neurodevelopmental conditions) and mothers (reduced risk of postpartum depression, ovarian and breast cancer, and type 2 diabetes) (Eidelman et al., 2012; Ip et al., 2007).

               We also removed the mention of obesity from the introduction (page 2, line 51):

Breastfeeding in tribal communities is particularly important as chronic health conditions such as diabetes disproportionately impact American Indian populations compared to other races and ethnicities (Fagot-Campagna et al., 2000; Kuperberg & Evers, 2007; Warne & Wescott, 2019).

  1. A total of 2136 mothers missed data on the question and were excluded. Will more than half the number of excluded population lead to result deviation? The authors needs to explain it.

In general, the women excluded from the analysis had a high degree of missingness for most of the variables examined below. Education Level and Race were the only 2 variables examined that had less than 93% missing, and both Education Level (1.5% missing) and Race (0% missing) were nearly complete or complete.

Given the high degree of missingness on key demographic and health variables among the excluded women, it is difficult to discern how the excluded women may be different than the women included in the analysis. However, given that the excluded participants are majority American Indian (62.50%) and a majority have Equal to or Less than High School Education (53.70%), the women excluded from analysis may be at high risk for late prenatal care not initiating breastfeeding or having short breastfeeding duration. While this look at the excluded women is limited due to the high degree of missingness overall, Race and Education Level suggest we excluded high risk women from the analysis, which leads us to suggest our findings in the analytic sample may be biased towards the null, thus are a conservative representation of the disparities in breastfeeding.

The original version of the manuscript included this text in the Methods section (page 4, line 158):

Of the women excluded from the study, majority were American Indian (62.50%), had an education level below high school (53.70%), and were not married (69.71%). After applying survey weights, the analytic sample was 29,845 women, thus each ND PRAMS participant represents approximately 13 women, on average, that recently gave birth in ND.

We also revised the limitations section to further clarify the implications of excluding high risk women from our sample (page 13, line 424):

Second, those excluded from analysis for missing variables were 62.5% AI/AN and had an education level below high school (53.70%), thus likely being high risk for not initiating breastfeeding and having short breastfeeding duration, as well as being high risk for experience of interpersonal violence. Thus,  the estimated disparity in breastfeeding outcomes, and the observed lack of effect attributed to interpersonal violence, can be considered conservative.

  1. The authors should supplement the inclusion flow chart to make the inclusion steps clearer.

               We have included the following flow chart as Figure 1 in the revised manuscript.

Figure 1. Study inclusion/exclusion flow chart

5.About statistics, the authors should supplement the P value between the two groups in Table 1 and Table 2 (different letter superscripts can be used). About OR value in the article, please add specific P value in corresponding description and table

We have added the p-value as superscript on both Table 1 and Table 2. We have also added a superscript for statistically significant p-values on the regression results (Tables 3-5).

The p-values were not added to the text of the manuscript as we already have the confidence intervals included in the text and in the tables. Both are measures of statistical significance, yet confidence intervals provide more information about the precision of the estimate. Solely relying on p-values for regression results is overly simplistic (being a dichotomous yes/no approach) and leads to poor understanding of whether or not the effect exists in the population under study. Here are links for further information on this concept:

https://www.ncbi.nlm.nih.gov/pmc/articles/PMC5738950/

https://www.tandfonline.com/doi/full/10.1080/00031305.2016.1154108

Reviewer 2 Report

The study conducted in the paper shed the light on an crucial issue, which is the low rate of breastfeeding among American Indian women. I have some comments:

1- Authors should explain the relation between violence (in all its kinds) and decreasing the rate of breastfeeding. kindly find here some references

https://journals.sagepub.com/doi/10.1177/1524838007304406

https://www.news-medical.net/news/20191001/Domestic-violence-may-affect-mothers-breastfeeding-practices-in-developing-countries.aspx

2- Authors should explain also the relation between racism and the low rate of breastfeeding

3- The education and the low income might affect the low rate of breastfeeding among the AI women. it can be seen from the data that there is a high percentage of low income and low educational attainment.

4- It is not very clear for me what is the main reason behind the low rate of breastfeeding among the AI women. it is important to identify it in order to deal with it and to increase the rate of breastfeeding among AI women.

Author Response

Reviewer #2:

The study conducted in the paper shed the light on an crucial issue, which is the low rate of breastfeeding among American Indian women. I have some comments:

1- Authors should explain the relation between violence (in all its kinds) and decreasing the rate of breastfeeding. kindly find here some references

https://journals.sagepub.com/doi/10.1177/1524838007304406

https://www.news-medical.net/news/20191001/Domestic-violence-may-affect-mothers-breastfeeding-practices-in-developing-countries.aspx

               We have included the following text in the introduction section on page 2, line 75:

Types of interpersonal violence include childhood maltreatment, family and intimate partner violence (IPV), and community violence, which occurs among non-related individuals (Mercy et al., 2017). All forms of interpersonal violence have been linked to poorer breastfeeding outcomes including less initiation and decreased duration in spite of violence survivors often having greater intentions to begin breastfeeding (Caleyachetty et al., 2019; Channell Doig et al., 2020; Normann et al., 2020). It appears the stress of being mistreated (Kendall-Tackett, 2007) interferes biologically and psychologically with breastfeeding. Even for those not currently being abused, qualitative research suggests that for childhood sexual abuse survivors, breastfeeding can activate body shame, feelings of lack of control of their bodies, and concerns around sexuality (Coles, 2009).

We have included the following text in the discussion section on page 11, line 324:

As noted in the Normann et al. 2020 systematic review, interpersonal violence is generally associated with poorer breastfeeding outcomes. Yet, the relationship between interpersonal violence and breastfeeding is complex and accounting for different covariates changes findings.

2- Authors should explain also the relation between racism and the low rate of breastfeeding

This comment is well received. In the discussion section as originally submitted, we describe how colonization and subsequent federal policy (like the Save the Babies campaign) have had disastrous impact on breastfeeding among American Indian populations. What we did not do originally is call out these actions for what they are – structural racism. Call these actions structural racism has now been included in the Discussion section on page 12, line 324:

The impact structural racism, in the form of colonization and subsequent federal policies, has had on breastfeeding behavior among American Indian populations is an additional barrier that is difficult to measure.

We also made revisions on page 12, line 380 to further clarify these programs were an attempt to discourage American Indian cultural and tradition at a policy level:

Field staff on many reservations began to discourage breastfeeding, which they presented traditional breastfeeding practices of American Indian communities as barbaric and backwards, and promoted cow’s milk as a more scientifically attractive mode of feeding infants (Theobald, B., 2019).

3- The education and the low income might affect the low rate of breastfeeding among the AI women. it can be seen from the data that there is a high percentage of low income and low educational attainment.

In order to determine if income or education may have a modifying effect on our results, we ran 2 additional versions of our regression models: 1) By excluding income from the fully adjusted model; 2) excluding education from the fully adjusted model. As seen in both tables below, excluding either income or education did not result in qualitatively different results. We have added these new analyses as sensitivity analyses to the paper, and the tables will be presented as supplemental files. We have added the following text in the methods section on page 5, line 216:

To determine if income or education may modify potential racial/ethnic disparities in breastfeeding, we performed sensitivity analyses by fitting two additional regression models for each breastfeeding outcome: 1) Excluding education from the fully adjusted model; 2) excluding income from the fully adjusted model.

We added the following to the results section on page 11, line 311:

Results of the sensitivity analyses do not differ from our main analyses, suggesting income and education do not greatly influence observed racial/ethnic disparities in breastfeeding (Supplemental tables 1-6).

Supplemental Table 1. Education variable removed from main breastfeeding initiation analyses.

Breastfeeding Initiation

OR (95% CI)

American Indian

Other Racial Identities

White

Violence Variable Estimate

Crude Model

0.24

(0.18, 0.31)

0.87

(0.49, 1.52)

Ref.

-

Sociodemographic Model

0.55

(0.37, 82)

1.09

(0.58, 2.06)

Ref.

-

Violence Before Pregnancy

Any Interpersonal Violence

0.54

(0.37, 0.81)

1.11

(0.59, 2.08)

Ref.

0.36

(0.10, 1.28)

Husband/Partner

0.55

(0.37, 0.82)

1.09

(0.58, 2.06)

Ref.

0.95

(0.37, 2.42)

Ex- Husband/Partner

0.55

(0.37, 0.82)

1.09

((0.58, 2.06)

Ref.

0.97

(0.35, 2.66)

Other Family

0.56

(0.37, 0.83)

1.10

(0.58, 2.06)

Ref.

0.65

(0.22, 1.89)

Other

0.55

(0.37, 0.81)

1.09

(0.58, 2.06)

Ref.

0.46

(0.18, 1.20)

Violence During Pregnancy

Any Interpersonal Violence

0.55

(0.37, 0.82)

1.09

(0.58, 2.07)

Ref.

0.66

(0.18, 2.34)

Husband/Partner

0.54

(0.36, 0.81)

1.08

(0.57, 2.05)

Ref.

1.34

(0.52, 3.47)

Ex- Husband/Partner

0.55

(0.37, 0.81)

1.09

(0.58, 2.06)

Ref.

1.41

(0.46, 4.37)

Other Family

0.55

(0.37, 0.82)

1.09

(0.58, 2.06)

Ref.

0.85

(0.33, 2.19)

Other

0.55

(0.37, 0.81)

1.09

(0.58, 2.06)

Ref.

1.23

(0.42, 3.59)

Adjusted for race, income, insurance before pregnancy, maternal age, substance use, marital status, pregnancy intention, chronic disease, overweight/obese, WIC, Kotelchuck, and ACEs.

Supplemental Table 2. Income variable removed from main breastfeeding initiation analyses.

Breastfeeding Initiation

OR (95% CI)

American Indian

Other Racial Identities

White

Violence Variable Estimate

Crude Model

0.24

(0.18, 0.31)

0.87

(0.49, 1.52)

Ref.

-

Sociodemographic Model

0.52

(0.35, 0.78)

1.02

(0.54, 1.93)

Ref.

-

Violence Before Pregnancy

Any Interpersonal Violence

0.52

(0.35, 0.77)

1.04

(0.55, 1.97)

Ref.

0.38

(0.10, 1.36)

Husband/Partner

0.52

(0.35, 0.78)

1.02

(0.54, 1.93)

Ref.

0.94

(0.37, 2.36)

Ex- Husband/Partner

0.52

(0.35, 0.78)

1.02

(0.54, 1.93)

Ref.

0.94

(0.35, 2.47)

Other Family

0.53

(0.35, 0.79)

1.03

(0.54, 1.94)

Ref.

0.67

(0.23, 1.92)

Other

0.52

(0.35, 0.78)

1.03

(0.54, 1.94)

Ref.

0.48

(0.8, 1.31)

Violence During Pregnancy

Any Interpersonal Violence

0.52

(0.35, 0.78)

1.02

(0.54, 1.94)

Ref.

0.77

(0.20, 2.85)

Husband/Partner

0.52

(0.34, 0.77)

1.02

(0.54, 1.92)

Ref.

1.37

(0.53, 3.50)

Ex- Husband/Partner

0.52

(0.35, 0.78)

1.02

(0.54, 1.93)

Ref.

1.30

(0.45, 3.69)

Other Family

0.52

(0.35, 0.78)

1.02

(0.54, 1.93)

Ref.

0.86

(0.34, 2.21)

Other

0.52

(0.35, 0.78)

1.02

(0.54, 1.93)

Ref.

1.26

(0.43, 3.71)

Adjusted for race, insurance before pregnancy, maternal age, education, substance use, marital status, pregnancy intention, chronic disease, overweight/obese, WIC, Kotelchuck, and ACEs.

Supplemental Table 3. Education variable removed from main 2 month breastfeeding duration analyses

Breastfeeding Duration – 2 Months

OR (95% CI)

American Indian

Other Racial Identities

White

Violence Variable Estimate

Crude Model

0.30

(0.25, 0.36)

1.08

(0.75, 1.57)

Ref.

-

Sociodemographic Model

0.70

(0.53, 0.93)

1.23

(0.76, 1.99)

Ref.

-

Violence Before Pregnancy

Any Interpersonal Violence

0.70

(0.52, 0.93)

1.25

(0.77, 2.02)

Ref.

0.35

(0.07, 1.56)

Husband/Partner

0.72

(0.54, 0.96)

1.24

(0.77, 2.01)

Ref.

0.60

(0.23, 1.53)

Ex- Husband/Partner

0.70

(0.53, 0.94)

1.24

(0.77, 2.00)

Ref.

0.69

(0.30, 1.60)

Other Family

0.72

(0.54, 0.96)

1.24

(0.77, 2.01)

Ref.

0.44

(0.13, 1.41)

Other

0.70

(0.53, 0.93)

1.24

(0.77, 1.99)

Ref.

0.47

(0.18, 1.18)

Violence During Pregnancy

Any Interpersonal Violence

0.71

(0.53, 0.94)

1.25

(0.77, 2.03)

Ref.

0.22

(0.05, 1.02)

Husband/Partner

0.72

(0.54, 0.96)

1.25

(0.78, 2.03)

Ref.

0.50

(0.20, 1.24)

Ex- Husband/Partner

0.71

(0.53, 0.94)

1.24

(0.77, 2.00)

Ref.

0.47

(0.18, 1.22)

Other Family

0.72

(0.54, 0.96)

1.24

(0.77, 2.00)

Ref.

0.50

(0.16, 1.53)

Other

0.70

(0.53, 0.94)

1.23

(0.76, 1.99)

Ref.

0.60

(0.22, 1.58)

Adjusted for race, income, insurance before pregnancy, maternal age, substance use, marital status, pregnancy intention, chronic disease, overweight/obese, WIC, Kotelchuck, and ACEs.

Supplemental Table 4. Income variable removed from main 2 month breastfeeding duration analyses

Breastfeeding Duration – 2 Months

OR (95% CI)

American Indian

Other Racial Identities

White

Violence Variable Estimate

Crude Model

0.30

(0.25, 0.36)

1.08

(0.75, 1.57)

Ref.

-

Sociodemographic Model

0.68

(0.51, 0.90)

1.18

(0.73, 1.89)

Ref.

-

Violence Before Pregnancy

Any Interpersonal Violence

0.68

(0.51, 0.90)

1.19

(0.74, 1.92)

Ref.

0.35

(0.08, 1.52)

Husband/Partner

0.70

(0.52, 0.93)

1.19

(0.74, 1.91)

Ref.

0.59

(0.23, 1.48)

Ex- Husband/Partner

0.68

(0.51, 0.90)

1.18

(0.74, 1.91)

Ref.

0.68

0.28, 1.59)

Other Family

0.69

(0.52, 0.92)

1.19

0.74, 1.91)

Ref.

0.44

(0.14, 1.39)

Other

0.68

(0.51, 0.90)

1.18

(0.73, 1.90)

Ref.

0.47

(0.18, 1.20)

Violence During Pregnancy

Any Interpersonal Violence

0.68

(0.51, 0.91)

1.19

(0.74, 1.92)

Ref.

0.23

(0.05, 1.09)

Husband/Partner

0.69

(0.52, 0.92)

1.19

(0.74, 1.92)

Ref.

0.50

(0.20, 1.25)

Ex- Husband/Partner

0.69

(0.52, 0.91)

1.18

(0.74, 1.90)

Ref.

0.44

(0.17, 1.14)

Other Family

0.69

(0.52, 0.92)

1.18

(0.74, 1.90)

Ref.

0.49

(0.16, 1.49)

Other

0.68

(0.51, 0.90)

1.18

(0.73, 1.89)

Ref.

0.59

(0.22, 1.55)

Adjusted for race, insurance before pregnancy, maternal age, education, substance use, marital status, pregnancy intention, chronic disease, overweight/obese, WIC, Kotelchuck, and ACEs.

Supplemental Table 5. Education variable removed from main 6 month breastfeeding duration analyses

Breastfeeding Duration – 6 Months

OR (95% CI)

American Indian

Other Racial Identities

White

Violence Variable Estimate

Crude Model

0.24

(0.20, 0.29)

0.87

(0.62, 1.23)

Ref.

-

Sociodemographic Model

0.62

(0.46, 0.83)

0.88

(0.56, 1.36)

Ref.

-

Violence Before Pregnancy

Any Interpersonal Violence

0.62

(0.46, 0.83)

0.88

(0.57, 1.37)

Ref.

0.58

(0.11, 3.00)

Husband/Partner

0.63

(0.47, 0.84)

0.88

(0.56, 1.37)

Ref.

0.69

(0.24, 1.98)

Ex- Husband/Partner

0.62

(0.46, 0.83)

0.88

(0.56, 1.37)

Ref.

0.47

(0.18, 1.20)

Other Family

0.63

(0.47, 1.37)

0.88

(0.57, 1.37)

Ref.

0.47

(0.12, 1.76)

Other

0.62

(0.46, 0.83)

0.88

(0.56, 1.36)

Ref.

0.69

(0.25, 1.93)

Violence During Pregnancy

Any Interpersonal Violence

0.62

(0.47, 0.83)

0.89

(0.57, 1.38)

Ref.

0.33

(0.06, 1.72)

Husband/Partner

0.63

(0.47, 0.84)

0.89

(0.57, 1.38)

Ref.

0.54

(0.20, 1.43)

Ex- Husband/Partner

0.62

(0.47, 0.83)

0.88

(0.56, 1.36)

Ref.

0.74

(0.26, 2.11)

Other Family

0.63

(0.47, 0.84)

0.88

(0.56, 1.37)

Ref.

0.70

(0.20, 2.45)

Other

0.62

(0.47, 0.83)

0.88

(0.56, 1.36)

Ref.

0.65

(0.23, 1.81)

Adjusted for race, income, insurance before pregnancy, maternal age, substance use, marital status, pregnancy intention, chronic disease, overweight/obese, WIC, Kotelchuck, and ACEs.

Supplemental Table 6. Income variable removed from main 6 month breastfeeding duration analyses

Breastfeeding Duration – 6 Months

OR (95% CI)

American Indian

Other Racial Identities

White

Violence Variable Estimate

Crude Model

0.24

(0.20, 0.29)

0.87

(0.62, 1.23)

Ref.

-

Sociodemographic Model

0.58

(0.44, 0.77)

0.79

(0.51, 1.22)

Ref.

-

Violence Before Pregnancy

Any Interpersonal Violence

0.58

(0.43, 0.77)

0.79

(0.51, 1.23)

Ref.

0.58

(0.12, 2.80)

Husband/Partner

0.59

(0.44, 0.78)

0.79

(0.51, 1.23)

Ref.

0.67

(0.24, 1.85)

Ex- Husband/Partner

0.58

(0.43, 0.77)

0.79

(0.51, 1.23)

Ref.

0.45

(0.18, 1.13)

Other Family

0.59

(0.44, 0.79)

0.79

(0.51, 1.23)

Ref.

0.46

(0.13, 1.65)

Other

0.58

(0.43, 0.77)

0.79

(0.51, 1.22)

Ref.

0.69

(0.24, 1.92)

Violence During Pregnancy

Any Interpersonal Violence

0.58

(0.44, 0.77)

0.80

(0.51, 1.23)

Ref.

0.36

(0.06, 1.90)

Husband/Partner

0.59

(0.44, 0.78)

0.80

(0.52, 1.24)

Ref.

0.54

(0.20, 1.45)

Ex- Husband/Partner

0.58

(0.44, 0.77)

0.79

(0.51, 1.22)

Ref.

0.67

(0.24, 1.87)

Other Family

0.59

(0.44, 0.78)

0.79

(0.51, 1.22)

Ref.

0.69

(0.21, 2.32)

Adjusted for race, insurance before pregnancy, maternal age, education, substance use, marital status, pregnancy intention, chronic disease, overweight/obese, WIC, Kotelchuck, and ACEs.

4- It is not very clear for me what is the main reason behind the low rate of breastfeeding among the AI women. it is important to identify it in order to deal with it and to increase the rate of breastfeeding among AI women.

               Yes, this is one of the most important challenges we face in addressing Maternal and Child Health disparities among American Indian women. We identified interpersonal violence as a potential variable that may account for the persistent disparity in breastfeeding, and were surprised at the little to no effect it had on the observed disparities in PRAMS data. We have a paper under review at a different journal in which we saw a similar lack of effect of interpersonal violence on access to prenatal care among American Indian women. Due to our (thus far) lack of explanation of these key disparities, we have hypothesized that there are structural issues at play (such as structural racism, historic trauma) as addressed in the Discussion section. We are in the midst of primary data collection to better answer some of our hypothesized questions in order to (hopefully) lead to well-informed public health approaches to improve Maternal and Child Health outcomes among American Indian populations.

Reviewer 3 Report

The researchers explore whether interpersonal violence partially explains racial/ethnic disparities in breastfeeding in North Dakota.

Although the variable of greatest interest does not seem to correlate with breastfeeding, this study can be considered an interesting contribution to the current debate on the socio-cultural and health policy aspects that characterize women's health in the perinatal period in native North American populations.

However, in order to be considered for possible publication in IJERPH, the present version of the manuscript needs some small additions and clarifications.

With what criteria were the questions relating to Breastfeeding and Interpersonal Violence formulated? Are there previous studies that have used this same methodology with single questions in populations of non-Indigenous women as well?

Discussion, limitation section line 322

It should be noted that using only one yes/no question on breastfeeding can be considered a limitation of the study. Additionally, there is no reference to the collection of data concerning exclusive and mixed breastfeeding or alternate nutrition, nor the relative duration of each type of feeding offered to the newborn. This omission can also be considered a limitation of the study. 

Author Response

Reviewer #3:

The researchers explore whether interpersonal violence partially explains racial/ethnic disparities in breastfeeding in North Dakota.

Although the variable of greatest interest does not seem to correlate with breastfeeding, this study can be considered an interesting contribution to the current debate on the socio-cultural and health policy aspects that characterize women's health in the perinatal period in native North American populations.

However, in order to be considered for possible publication in IJERPH, the present version of the manuscript needs some small additions and clarifications.

With what criteria were the questions relating to Breastfeeding and Interpersonal Violence formulated? Are there previous studies that have used this same methodology with single questions in populations of non-Indigenous women as well?

The breastfeeding and IPV questions in the ND PRAMS survey were developed by the Centers of Disease Control and Prevention and made available as part of a "Standard List" of questions that states can choose to include in their survey. As with all PRAMS questions, since 1987, these questions have been tested among diverse populations and have undergone the standard flow assessment that is part of the CDC PRAMS Question Development Protocol.  The following reference explains the methodology: Shulman, Holly B., et al. "The pregnancy risk assessment monitoring system (PRAMS): overview of design and methodology." American journal of public health 108.10 (2018): 1305-1313.

We have included the following text in the Methods section on page 3, line 138:

PRAMS survey questions were developed by the CDC and made available as part of a “standard list” that states choose to include in their survey. These questions have been tested among diverse populations and have undergone assessment as part of CDC protocol. Survey methodology is the same for all racial/ethnic groups sampled.

Discussion, limitation section line 322

It should be noted that using only one yes/no question on breastfeeding can be considered a limitation of the study. Additionally, there is no reference to the collection of data concerning exclusive and mixed breastfeeding or alternate nutrition, nor the relative duration of each type of feeding offered to the newborn. This omission can also be considered a limitation of the study. 

               Thank you for this suggestion. We have added the following text in the limitations section (page 13, line 429):

Third, PRAMS only collects one yes/no question on breastfeeding initiation, and does not collect data on exclusive breastfeeding or mixed feedings. Thus, our observations do not apply to mixed feeding, alternative nutrition, or introduction of solid foods for infants.

Reviewer 4 Report

The manuscript "Exposure to Interpersonal Violence and Racial Disparities in Breastfeeding in North Dakota (ND PRAMS 2017-2019)" analyzes the extent to which violence in certain races residing in the North Dakota area affects breastfeeding initiation or its early abandonment.
The article is interesting, the results provide answers to the basic question, however I would suggest some refinements.
I suggest improving the abstract by specifying, in very few words, what the authors explain best in the introduction regarding the reasons for not breastfeeding in 16% of women. Similarly, a brief mention of why violence could be one of the reasons would be in order.
On the contrary, the questions used for data collection should not be explained in detail in the abstract, it should only be specified if, for example, a validated questionnaire was used or a questionnaire was defined by the researchers specifically for the occasion, if the questions were closed or open and the like.
Before line 45 it would be necessary to add why breastfeeding is important for the prevention of obesity and diabetes.
Materials and methods: this part needs to be improved. First of all it would seem that the authors did not collect the socio-demographic, clinical data, etc. while such data are available and are presented in the results. Therefore, all the data that has been collected and how they were collected should be listed. In this part, the questionnaire that was used must be explained well (if it has been explained in previous publications, it must be specified where): type of questions, scales, etc. Kotelchuck needs to be explained.
Table 2 - in the title I would use the terms absolute and relative frequencies.
Conclusions: must be rewritten in order to reflect the results of the manuscript and only briefly suggest how to proceed in the future (how to apply the results of the study in practice).

Author Response

Reviewer #4:

The manuscript "Exposure to Interpersonal Violence and Racial Disparities in Breastfeeding in North Dakota (ND PRAMS 2017-2019)" analyzes the extent to which violence in certain races residing in the North Dakota area affects breastfeeding initiation or its early abandonment.

The article is interesting, the results provide answers to the basic question, however I would suggest some refinements.

I suggest improving the abstract by specifying, in very few words, what the authors explain best in the introduction regarding the reasons for not breastfeeding in 16% of women. Similarly, a brief mention of why violence could be one of the reasons would be in order.

We have added the following text to the abstract regarding violence and breastfeeding:

Stress associated with interpersonal violence may interfere with processes important to breastfeeding.

On the contrary, the questions used for data collection should not be explained in detail in the abstract, it should only be specified if, for example, a validated questionnaire was used or a questionnaire was defined by the researchers specifically for the occasion, if the questions were closed or open and the like.

Thank you for this suggestion. However, for clarity to future readers, we have decided to keep the specific questions listed in the abstract. We did add in the following language to better describe the validity of these specific questions:

Breastfeeding questions in PRAMS have been tested among diverse populations.

Before line 45 it would be necessary to add why breastfeeding is important for the prevention of obesity and diabetes.

Thank you for this comment. Upon additional review of the literature, we have removed the mention of obesity, as the evidence is weak that maternal breastfeeding reduces risk for maternal obesity later in life. However, the evidence is sufficient to suggest that maternal breastfeeding reduces maternal risk for type 2 diabetes in the future. We have revise the text in the introduction to explain that breastfeeding is associated with lower risk of type 2 diabetes after pregnancy (page 1, line 44):

Breastfeeding provides an array of health benefits for both babies (reduced risk of lower respiratory tract infections, sudden infant death syndrome, and neurodevelopmental conditions) and mothers (reduced risk of postpartum depression, ovarian and breast cancer, and type 2 diabetes) (Eidelman et al., 2012; Ip et al., 2007).

               We also removed the mention of obesity from the introduction (page 2, line 50):

Breastfeeding in tribal communities is particularly important as chronic health conditions such as diabetes disproportionately impact American Indian populations compared to other races and ethnicities (Fagot-Campagna et al., 2000; Kuperberg & Evers, 2007; Warne & Wescott, 2019).

Materials and methods: this part needs to be improved. First of all it would seem that the authors did not collect the socio-demographic, clinical data, etc. while such data are available and are presented in the results. Therefore, all the data that has been collected and how they were collected should be listed. In this part, the questionnaire that was used must be explained well (if it has been explained in previous publications, it must be specified where): type of questions, scales, etc. Kotelchuck needs to be explained.

To provide better explanation of the survey tool, we have included the following text in the methods section on page 3, line 134:

At the time of data collection, the ND PRAMS survey consisted of 103 multiple choice and fill-in-the-blank questions regarding sociodemographic factors, maternal behaviors and experiences, and clinical outcomes. Data from birth certificates is linked to ND PRAMS data to supplement the survey results.

As for further description of all scales, indices, etc, we are limited by word count to fully describe all procedures for all variables. However, we have added additional information about the source of each variable, as well as included a citation for the Kotelchuck index. The Kotelchuck index is a commonly used measure of prenatal care adequacy, and based on the number of prenatal visits completed based on the beginning of prenatal care until delivery. This section is on page 5, line 203 of the Methods:

Models were adjusted for maternal age (less than 20 years old, 21-35 years old, and greater than 35 years old; self-report), income (less than $48,000 and greater than $48,000; self-report), education (less than high school and greater than high school; derived from birth certificate), participation in the Women, Infants, and Children Nutrition (WIC; derived from birth certificate) program (yes, no), insurance prior to pregnancy (yes, no; self-report), pregnancy intention (yes, no; self-report), Kotelchuck Adequacy of Prenatal Care Utilization (APNCU; derived from birth certificate data) index (inadequate, intermediate, adequate, adequate plus), Body Mass Index (less than 25, greater than 25; self-report), substance use during pregnancy (cigarette, marijuana, and e-cigarette use; self-report), marital status (yes, no; derived from birth certificate), presence of chronic illness and conditions (diabetes, high blood pressure, and depression; self-report), and adverse childhood experiences (ACEs) (less than 4, 4 or more; self-report). Covariates were informed by prior literature. Statistical analyses were conducted with SAS OnDemand for Academics (SAS Institute Inc., Cary, NC, USA) using proc survey commands and survey weights to account for complex survey design and nonresponse.

Table 2 - in the title I would use the terms absolute and relative frequencies.

We have updated the title for Table 2 as follows (page 7, line 247):

Table 2. Weighted Relative (and absolute) frequency of violence exposure by race/ethnicity in ND PRAMS

Conclusions: must be rewritten in order to reflect the results of the manuscript and only briefly suggest how to proceed in the future (how to apply the results of the study in practice).

We have rewritten the conclusion to reduce text regarding future efforts (page 14, line 442):

Our results highlight that breastfeeding disparities exist among American Indian women and prevalent risk factors such as interpersonal violence do not explain observed disparities.   Culturally responsive interventions (e.g. breastfeeding education and lactation support) are needed to improve breastfeeding outcomes among American Indian mothers, yet research supporting culturally responsive interventions is sparse within AI populations. Further research should offer clarification to unique socioeconomic circumstances, such as the women’s role as a breadwinner, and its impact on breastfeeding disparities. As a historically systemically oppressed population, in which health policies have diminished cultural practices in the past, ethical research practices; including data sovereignty, data ownership, and publications should be at the respective tribe’s discretion in relation to any future research.

Reviewer 5 Report

Kanichy and colleagues assessed the race disparities in breastfeeding in ND and evaluated the impact of sociodemographic and clinical characteristics and violence exposure. Along with previous evidence well-acknowledged in this manuscript, the lower breastfeeding rate among American Indian women is prominent as compared to other racial groups in ND using the real-world data from the ND Pregnancy Risk Assessment Monitoring System. Regarding the assessment of interpersonal violence, the reviewer has several questions and concerns about the findings and conclusion of the lack of impact of violence exposure. 

#1) Line 128:  First, the high nonresponse rate to breastfeeding status in AI women (~63%) is a concern despite it being acknowledged by the authors. The limitation might need more elaboration and clarification regarding the nature of data collection if relevant. The reviewer wondered to what extent the population could be represented by the final sample. 

#2) Line 126: “Did you ever breastfeed or pump breast milk to feed your new baby, even for a short period of time” appeared to be the only question applied as an inclusion criterion. The reviewer wondered about the levels of missingness across other violence- and breastfeeding-related questions and whether they impacted the statistical power and results. 

#3) Line 151: The inclusion of “any exposure to violence” was unclear to the reviewer. Did it refer to any above types of violence or other types not included in other categories? In Table 2, the number of “Yes” responses to “Any exposure to violence” appeared to be much smaller than other types of violence exposure. Please clarify. Besides, the Yes response to each type of violence exposure is quite small. The reviewer wondered whether it attenuated the statistical power for association analyses with race/breastfeeding. The authors could create a composite variable including all types of violence exposure. 

#4) The violence of exposure variables appeared to be solely analyzed in the multiple regressions of racial groups and breastfeeding outcomes. The reviewer wondered about the direct relationships between violence exposure and breastfeeding outcomes. 

Besides the above questions, the reviewer had the following additional comments for the authors to consider to further improve this work: 

#5) What were the rationales to examine the selected sociodemographic and clinical characteristics in association with breastfeeding? Were any feature selection procedures performed prior to the formal regression analysis? Or supporting evidence from previous studies? 

6#) Are there any specific known cultural beliefs surrounding breastfeeding that are relevant to the lower rate in AI as compared to other racial groups? Commenting on this point would be helpful for the audience to better understand this public health issue.

Author Response

Reviewer #5:

Kanichy and colleagues assessed the race disparities in breastfeeding in ND and evaluated the impact of sociodemographic and clinical characteristics and violence exposure. Along with previous evidence well-acknowledged in this manuscript, the lower breastfeeding rate among American Indian women is prominent as compared to other racial groups in ND using the real-world data from the ND Pregnancy Risk Assessment Monitoring System. Regarding the assessment of interpersonal violence, the reviewer has several questions and concerns about the findings and conclusion of the lack of impact of violence exposure. 

#1) Line 128:  First, the high nonresponse rate to breastfeeding status in AI women (~63%) is a concern despite it being acknowledged by the authors. The limitation might need more elaboration and clarification regarding the nature of data collection if relevant. The reviewer wondered to what extent the population could be represented by the final sample. 

In general, the women excluded from the analysis had a high degree of missingness for most of the variables examined below. Education Level and Race were the only 2 variables examined that had less than 93% missing, and both Education Level (1.5% missing) and Race (0% missing) were nearly complete or complete.

Given the high degree of missingness on key demographic and health variables among the excluded women, it is difficult to discern how the excluded women may be different than the women included in the analysis. However, given that the excluded participants are majority American Indian (62.50%) and a majority have Equal to or Less than High School Education (53.70%), the women excluded from analysis may be at high risk for late prenatal care not initiating breastfeeding or having short breastfeeding duration. While this look at the excluded women is limited due to the high degree of missingness overall, Race and Education Level suggest we excluded high risk women from the analysis, which leads us to suggest our findings in the analytic sample may be biased towards the null, thus are a conservative representation of the disparities in breastfeeding.

The original version of the manuscript included this text in the Methods section (page 4, line 158):

Of the women excluded from the study, majority were American Indian (62.50%), had an education level below high school (53.70%), and were not married (69.71%). After applying survey weights, the analytic sample was 29,845 women, thus each ND PRAMS participant represents approximately 13 women, on average, that recently gave birth in ND.

We also revised the limitations section to further clarify the implications of excluding high risk women from our sample (page 13, line 427):

Second, those excluded from analysis for missing variables were 62.5% AI/AN and had an education level below high school (53.70%), thus likely being high risk for not initiating breastfeeding and having short breastfeeding duration, as well as being high risk for experience of interpersonal violence. Thus,  the estimated disparity in breastfeeding outcomes, and the observed lack of effect attributed to interpersonal violence, can be considered conservative.

#2) Line 126: “Did you ever breastfeed or pump breast milk to feed your new baby, even for a short period of time” appeared to be the only question applied as an inclusion criterion. The reviewer wondered about the levels of missingness across other violence- and breastfeeding-related questions and whether they impacted the statistical power and results. 

We examined missingness in the violence and breastfeeding questions. Other than the breastfeeding initiation question, there was no missing on the breastfeeding duration questions, as only those women than initiated breastfeeding would answer those questions. We also did not observe any missing values on the interpersonal violence questions. Thus, reducing our sample based on the breastfeeding initiation question was necessary given the large amount of missingness for that variable, as noted by the reviewer.

#3) Line 151: The inclusion of “any exposure to violence” was unclear to the reviewer. Did it refer to any above types of violence or other types not included in other categories? In Table 2, the number of “Yes” responses to “Any exposure to violence” appeared to be much smaller than other types of violence exposure. Please clarify. Besides, the Yes response to each type of violence exposure is quite small. The reviewer wondered whether it attenuated the statistical power for association analyses with race/breastfeeding. The authors could create a composite variable including all types of violence exposure. 

               The “any exposure to violence” variable is a composite variable created by combining the responses to “husband/partner, ex-husband/ex-partner, another family member, or someone else” sources of violence. This was do such that if a PRAMS participant reported “yes” to any of the “husband/partner, ex-husband/ex-partner, another family member, or someone else” sources of violence then they were identified as “yes” to the ‘any exposure of violence’ variable. We have updated language on page 5, line 191 in the Methods section:

An “Any exposure to violence” variable was created if women reported “yes” to any of the exposure types (husband/partner, ex-husband/ex-partner, another family member, or someone else).

Thank you for checking our Table 2 closely. We mistakenly submitted and erroneous version of the table. We checked the coding and results for Table 2 and updated the text (on page 5, line 228) and table as needed. We also checked the coding for our regression models, and our models were correct as originally submitted.

experience higher rates of any interpersonal violence before pregnancy (11.68%, 2.75%, 1.54% respectively), and during pregnancy (7.16%, 1.70%, 1.72% respectively).

Table 2. Weighted Relative (and absolute) frequency of violence exposure by race/ethnicity in ND PRAMS.*

Overall

(n=29845)

American Indian

(n=2499)

White

(n=23585)

Other Racial Identities

(n=3760)

Any Violence Exposure Before Pregnancya

Yes

3.35 (1000)

11.68 (292)

2.75 (649)

1.54 (58)

No

96.65 (28845)

88.32 (2207)

97.24 (22936)

98.46 (3702)

Husband/Partnera

Yes

2.07 (618)

7.64 (191)

1.48 (349)

2.05 (77)

No

97.93 (29227)

92.36 (2308)

98.52 (23236)

97.95 (3683)

Ex-Husband/Partnera

Yes

2.12 (632)

5.60 (140)

1.69 (399)

2.45 (92)

No

97.88 (29213)

94.40 (2359)

98.31 (23186)

97.55 (3668)

Familya

Yes

1.50 (447)

5.76 (144)

0.94 (221)

2.15 (81)

No

98.50 (29398)

94.24 (2355)

99.06 (23364)

97.85 (3679)

Other

Yes

1.93 (575)

4.04 (101)

1.77 (418)

1.46 (55)

No

98.07 (29270)

95.96 (2398)

98.23 (23167)

98.54 (3705)

Any Violence Exposure During Pregnancy

Yes

2.17 (647)

7.16 (179)

1.70 (403)

1.72 (65)

No

97.83 (29198)

92.84 (2320)

98.30 (23182)

98.28 (3695)

Husband/Partnera

Yes

2.01 (601)

6.20 (155)

1.42 (336)

2.90 (109)

No

97.99 (29244)

93.80 (2344)

98.58 (23249)

97.10 (3651)

Ex-Husband/Partnera

Yes

1.45 (433)

4.32 (108)

1.12 (264)

1.59 (60)

No

98.55 (29412)

95.68 (2391)

98.88 (23321)

98.40 (3700)

Familya

Yes

1.47 (438)

5.80 (145)

0.94 (222)

1.84 (69)

No

98.53 (29407)

94.20 (2354)

99.06 (23363)

98.16 (3691)

Other

Yes

1.57 (468)

3.40 (85)

1.44 (340)

1.14 (43)

No

98.43 (29377)

96.60 (2414)

98.56 (23245)

98.86 (3717)

*P-values for racial/ethnic differences obtained with Rao-Scott chi square in proc surveyfreq

  1. p-value for racial/ethnic differences <.05

#4) The violence of exposure variables appeared to be solely analyzed in the multiple regressions of racial groups and breastfeeding outcomes. The reviewer wondered about the direct relationships between violence exposure and breastfeeding outcomes. 

Yes, the exposure variables were included in the regression models, with race/ethnicity as the main predictor variable of breastfeeding outcomes.  We have included the violence variable estimates in all our regression tables, and despite the relatively strong negative correlation between violence and all breastfeeding outcomes, we didn’t see a meaningful change in the observed racial/ethnic disparities in breastfeeding.

Below is a table of the crude relationship between the violence variables and the breastfeeding outcomes. We have decided not to include this table in the paper, as it doesn’t provide additional information compared to what is already included in the main regression tables. In short, whether in crude or fully adjusted models, experience of interpersonal violence is associated with low odds of breastfeeding.

Crude association between exposure to violence and breastfeeding duration

Breastfeeding Initiation

OR (95% CI)

2 months

OR (95% CI)

6 months

OR (95% CI)

Before Pregnancy

Any exposure

0.16

(0.05, 0.51)

0.19

(0.04, 0.69)

0.28

(0.08, 0.98)

Husband/Partner

0.36

(0.17, 0.81)

0.38

(0.17, 0.76)

0.41

(0.18, 0.88)

Ex-Husband/Partner

0.30

(0.13, 0.69)

0.29

(0.14, 0.62)

0.19

(0.08, 0.45)

Family

0.24

(0.10, 0.58)

0.28

(0.12, 0.64)

0.30

(0.12, 0.72)

Other

0.23

(0.10, 0.52)

0.27

(0.13, 0.60)

0.36

(0.17, 0.80)

During Pregnancy

Any exposure

0.25

(0.07, 0.84)

0.17

(0.05, 0.58)

0.25

(0.07, 0.83)

Husband/Partner

0.50

(0.21, 1.21)

0.40

(0.18, 0.83)

0.44

(0.20, 0.95)

Ex-Husband/Partner

0.36

(0.14, 0.96)

0.24

(0.10, 0.58)

0.33

(0.14, 0.80)

Family

0.30

(0.12, 0.73)

0.33

(0.15, 0.75)

0.43

(0.19, 0.98)

Other

0.45

(0.16, 1.25)

0.41

(0.19, 0.90)

0.47

(0.21, 1.04)

Besides the above questions, the reviewer had the following additional comments for the authors to consider to further improve this work: 

#5) What were the rationales to examine the selected sociodemographic and clinical characteristics in association with breastfeeding? Were any feature selection procedures performed prior to the formal regression analysis? Or supporting evidence from previous studies? 

               Covariates were identified based on prior studies of breastfeeding. We have updated the methods section on page 5, line 214 with the following:

               Covariates were informed by prior literature.

6#) Are there any specific known cultural beliefs surrounding breastfeeding that are relevant to the lower rate in AI as compared to other racial groups? Commenting on this point would be helpful for the audience to better understand this public health issue.

               We have included the following text in the Introduction (page 2, line 56):

Breastfeeding is a way Indigenous mothers can reclaim traditional practices by connecting with baby physically, emotionally, and spiritually. Furthermore, data suggest a strong association between prenatal use of traditional practices and breastfeeding at 6-months among Indigenous mothers in the Midwest region [9].

Round 2

Reviewer 1 Report

Thank you for addressing my comments with the updated manuscript. 

I believe the manuscript has been strengthened and recommend approval

Reviewer 5 Report

Thank you for taking my suggestions into consideration. I do not have further comments. It's my pleasure to review your submission.